# Electrical and magnetic anisotropies in van der Waals multiferroic CuCrP$_2$S$_6$

Xiaolei Wang [1] ✉, Zixuan Shang[1], Chen Zhang[2], Jiaqian Kang[3], Tao Liu[4], Xueyun Wang [3] ✉, Siliang Chen[5], Haoliang Liu [5] ✉, Wei Tang[6], Yu-Jia Zeng [6] ✉, Jianfeng Guo[7], Zhihai Cheng [7], Lei Liu[2], Dong Pan [2], Shucheng Tong[2], Bo Wu[8], Yiyang Xie [8], Guangcheng Wang[1], Jinxiang Deng[1], Tianrui Zhai [1], Hui-Xiong Deng [2], Jiawang Hong [3] & Jianhua Zhao[2]

Multiferroic materials have great potential in non-volatile devices for low-power and ultra-high density information storage, owing to their unique characteristic of coexisting ferroelectric and ferromagnetic orders. The effective manipulation of their intrinsic anisotropy makes it promising to control multiple degrees of the storage "medium". Here, we have discovered intriguing in-plane electrical and magnetic anisotropies in van der Waals (vdW) multiferroic CuCrP$_2$S$_6$. The uniaxial anisotropies of current rectifications, magnetic properties and magnon modes are demonstrated and manipulated by electric direction/polarity, temperature variation and magnetic field. More important, we have discovered the spin-flop transition corresponding to specific resonance modes, and determined the anisotropy parameters by consistent model fittings and theoretical calculations. Our work provides in-depth investigation and quantitative analysis of electrical and magnetic anisotropies with the same easy axis in vdW multiferroics, which will stimulate potential device applications of artificial bionic synapses, multi-terminal spintronic chips and magnetoelectric devices.

In the field of modern information storage, the physical properties could act as the storage "medium", such as electric polarization (dipole orientation) of ferroelectric materials[1,2], or spin polarization (magnetic vector) of magnetic materials[3,4]. It is worth noting that multiferroic materials simultaneously own both features and even their cross-coupling, which are promising to fabricate magnetoelectric storage devices, as the energy consumption is much lower than that of traditional semiconductor-based storage devices[5–8]. However, traditional

multiferroic materials are difficult to meet the development demand for potential applications due to the influence of size limit, interface effect, polarization origin, and reversal mechanism.

Compared to the traditional systems, two-dimensional (2D) van der Waals (vdW) materials exhibit obviously different and special physical properties with stable layered structures consisting of strong intralayer and weak interlayer forces[9–12]. In addition, 2D vdW materials are feasible to stack and integrate with various materials down to the

[1]Department of Physics and Optoelectronic Engineering, Faculty of Science, Beijing University of Technology, Beijing 100124, China. [2]State Key Laboratory of Superlattices and Microstructures, Institute of Semiconductors, Chinese Academy of Sciences, Beijing 100083, China. [3]School of Aerospace Engineering, Beijing Institute of Technology, Beijing 100081, China. [4]National Engineering Research Center of Electromagnetic Radiation Control Materials, University of Electronic Science and Technology of China, Chengdu 610054, China. [5]Guangdong Provincial Key Laboratory of Semiconductor, Optoelectronic Materials and Intelligent Photonic Systems, School of Science, Harbin Institute of Technology (Shenzhen), Shenzhen 518055, China. [6]Key Laboratory of Optoelectronic Devices and Systems of Ministry of Education and Guangdong Province, College of Physics and Optoelectronic Engineering, Shenzhen University, Shenzhen 518060, China. [7]Department of Physics and Beijing Key Laboratory of Opto-electronic Functional Materials & Micro-nano Devices, Renmin University of China, Beijing 100872, China. [8]Key Laboratory of Optoelectronics Technology Ministry of Education, Beijing University of Technology, Beijing 100124, China. ✉e-mail: xiaoleiwang@bjut.edu.cn; xueyun@bit.edu.cn; liuhaoliang@hit.edu.cn; yjzeng@szu.edu.cn

atomic-layer thickness, successfully avoiding size or interface issues[13–16]. Therefore, the investigation for intrinsic 2D vdW multi-ferroic materials has become one of the important directions for the development of next-generation information storage devices. The manipulation of electric and spin polarization in this type of material demonstrates obvious superiority for quantum information storage. Nevertheless, a desirable intrinsic 2D vdW multiferroic is rare, which calls for more research for discovering new materials.

Recently, the star compound $CuCrP_2S_6$ (CCPS), a type-II multi-ferroic material, has generated much excitement and is considered as a promising 2D vdW candidate owing to the coexistence of anti-ferroelectricity and antiferromagnetism, along with strong polarization-magnetization coupling[17–24]. CCPS exhibits alternating arrangement of Cu and Cr on honeycomb sublattice, which has a notable advantage of tunable ionic and spin characteristics for magnetoelectric storage devices[20,22]. The staggered displacement of the $Cu^+$ position away from the center of surrounding $S_6$ octahedra contributes to a antiferro-electric state, while the $Cr^{3+}$ sublattice gives rise to an antiferromagnetic phase below Néel temperature ($T_N$)[25,26]. These special features make CCPS a superior candidate for multiferroic application in functional 2D devices.

Systems with intrinsic in-plane anisotropy are of special interest and have great potential for realization of multi-terminal devices. Exploring the spontaneous intralayer electrical and magnetic aniso-tropies in a 2D vdW multiferroic material is of vital importance for fundamental investigation and storage device applications. Both electric dipoles and spin orders of CCPS stem from spin-orbit cou-pling, originating from crystal symmetry breaking[26–28]. However, its in-plane electrical and magnetic anisotropies have not been investigated thoroughly, and how the ionic migration interacts with the crystal lattice at nanoscale remain to be elucidated.

In this work, we show the evident electrical and magnetic ani-sotropies in the vdW layer of CCPS, which are dependent on crystal orientations and polarized directions. The in-plane electrical anisotropy is induced by the diffusion barriers along distinct axes, which causes different Cu ions migration efficiencies and current rectification behaviors. This discovery will stimulate potential device applications, such as artificial bionic synapses and multi-terminal spintronic devices. Moreover, we demonstrate the uniaxial magnetic anisotropy and spin-flop (SF) rotation of the Néel vector, with intra-layer ferromagnetic (FM) coupling and interlayer antiferromagnetic (AFM) coupling. Consistent results are obtained by antiferromagnetic resonance (AFMR) measurements, indicating the corresponding magnon modes and obtaining the numerical anisotropy parameters, which are supported by Landau–Lifshitz model fittings and first-principle calculations. The magnetic anisotropy, which determines the spin orientation, plays an important role in magnetic data storage and spintronic devices. Our work provides the fundamental under-standing and a good reference for future investigation in manipulat-ing electrical and magnetic anisotropies in vdW multiferroic CCPS, which could excite multi-functionalities for high-density non-volatile storage devices.

## Results

The single-crystal of CCPS, with monoclinic $Pc$ space group and stacked 2D layers connected by the vdW interaction, was grown by the chemical vapor transport method. It is known to be in the anti-ferroelectric (AFE) phase below the critical transition temperature $T_C \sim 145\,K$[17,23,26]. The Cu ions occupy the upper and lower position alternatively, resulting in absence of spontaneous macroscopic polarization. The side-view structure (interlayer $bc$-plane) in the AFE state is indicated in Fig. 1a, while the top-view structure (intralayer $ab$-plane) is shown in Fig. 1b. The triangular networks of CCPS include quasi-trigonal $CuS_3$, octahedral $CrS_6$, and $P_2S_6$ units. $Cu^+$ and $Cr^{3+}$ ions form a honeycomb lattice in an ordered way in the $ab$-plane. The $Cr^{3+}$

ions are almost centered in the vdW layer, whereas the $Cu^+$ ions are off-centered with a staggered displacement.

The X-ray diffraction (XRD) pattern on the surface of bulk CCPS indicates good single crystallographic orientation along the $c$ axis, as shown in Fig. 1c. The crystal bulk was formed as flexible platelet, which is captured in the inset. To confirm the good crystalline and high homo-geneity, the transmission electron microscope (TEM) was performed[29]. The TEM image of a well-prepared sample is characterized in Fig. 1d, with high-resolution capture of the local crystal structure in the inset. The simulated electron diffraction image of the monoclinic structure of CCPS along the $c$ axis (out of the $ab$-plane) is shown in Fig. 1e, which is in good agreement with that of our measurement in Fig. 1f. A little twist of the connected hexagon in Fig. 1f should come from the stretch of the CCPS thin flakes during ultrasonic treatment of sample preparation, which is not as regular as the simulated one in Fig. 1e. In the following, we focus on the electrical anisotropy by modulation of Cu cations within the vdW layer, and magnetic anisotropy through manipulating the spin orientation originating from the Cr site.

The piezoresponse force microscope (PFM) measurement was conducted on a CCPS crystal bulk (thick flake of ~3 μm thickness). The saturated and symmetric hysteresis loop of phase $vs$ DC bias voltage at low temperature was obtained, as shown in Fig. 2a. We observed the distinct 180° switching of ferroelectric polarization as the electric field is applied out-of-plane (along the $c$-axis). The structure diagram of emerged metastable ferroelectricity (FE) is represented in the upper inset of Fig. 2a, consisting of Cu atoms lying on the same side in a vdW layer. The out-of-plane electric dipole could originate from the AFE domain boundaries with unbalanced charges, which are related to defect-dipole polarization under the application of an electric field[12,24,25]. The measurement of magnetoelectric current $vs$ magnetic field has been also carried out to demonstrate strong magnetoelectric coupling in our CCPS crystal bulk, as shown in Supplementary Fig. 1. To investi-gate the electrical anisotropy and apply it to storage function[30], we fabricated the micro-devices and measured the current rectifications within the vdW layer of CCPS. We mechanically exfoliated the bulk CCPS and transferred thin flakes onto $SiO_2$/Si wafers by polydimethylsiloxane (PDMS). After the photolithography process, four Platinum (Pt) elec-trodes were deposited on top of the CCPS film, making the current channels along the recognized $a$ and $b$ axes of crystal structure, respectively. The microscope picture of a CCPS device and the scale-up configuration are illustrated in Fig. 2b, showing the electrical mea-surement design with identified $a$ and $b$ directions. The voltage was applied from one Pt electrode to the adjacent one, and different adja-cent electrode pairs could determine the different transport directions.

All the rectification behaviors in Fig. 2c–f were obtained on a 170-nm-thick CCPS nanoflake at room temperature, and the current-voltage ($I$–$V$) characteristics in the $ab$-plane were collected. After normalizing the hysteretic curves to current density $vs$ sweeping vol-tage ($J$–$V$), a clear electrical anisotropy is observed along the $a$ and $b$ directions after the same 10 V poling. The difference proportion of electrical anisotropy reaches a ratio of ~18 as applying +10 V poling for 3 min in Fig. 2c, while it decreases to a ratio of ~2 as increasing the time of poling voltage to 6 min or longer in Fig. 2d. When DC bias of 10 V is kept, the obvious difference of current density as a function of time along the $a$ and $b$ axes are observed in Fig. 2e. It is demonstrated that different rectifying characteristics are dependent on transport direc-tions and voltage polarities, and the easy conductive direction is along the $a$ axis.

After reversing the poling voltage to −10 V, the $J$–$V$ curve along the $a$ axis shows similar rectification behavior in the negative direction, as shown in Fig. 2f. It is indicated that the bipolar rectification behavior is achieved in a tunable single-phase memristor of CCPS. Other devices with different film thickness of CCPS (see Supplementary Fig. 2) were fabricated and measured to demonstrate the repeatability and relia-bility of our findings. The electrical property of our device originates

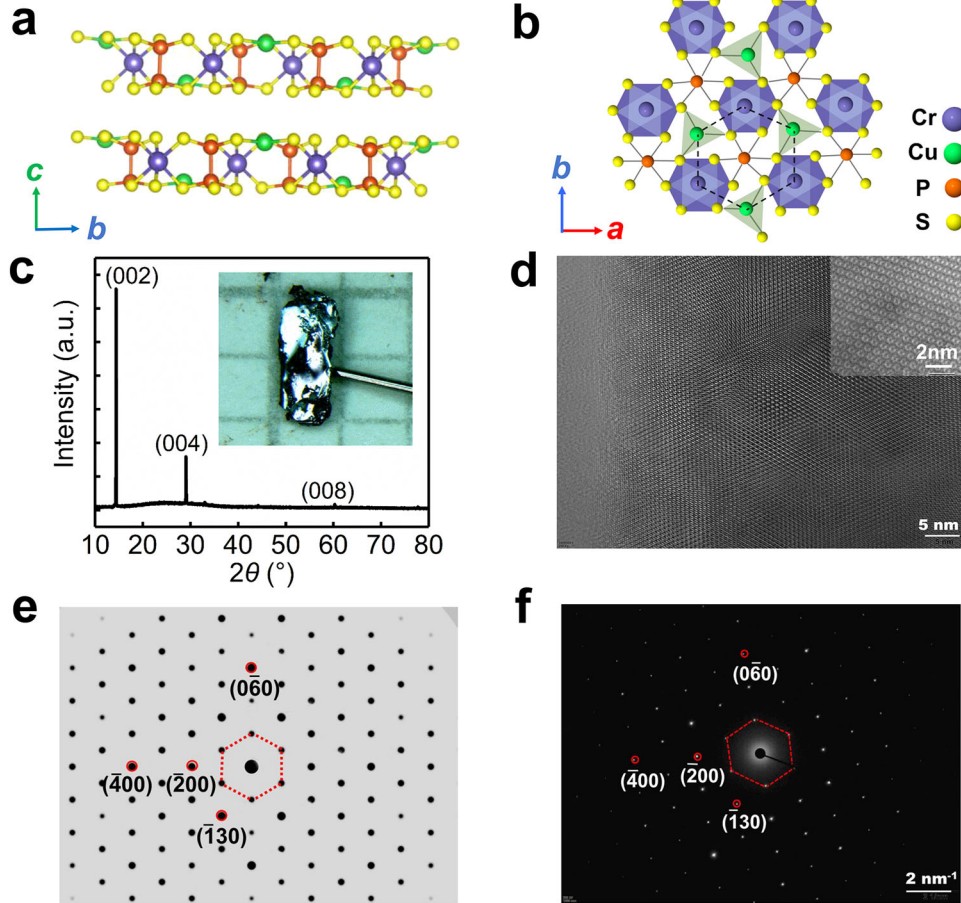

**Fig. 1 | Characterizations of vdW multiferroic CCPS single crystal. a** The side-view (interlayer *bc*-plane) and **b** the top-view (intralayer *ab*-plane) of CCPS atomic structure, Cu, Cr, P, and S ions are represented by green, blue, red, and yellow balls, respectively. **c** The XRD scan indicating (002), (004), and (008) peak reflections for the *ab*-plane of as-grown CCPS single crystal. The inset shows a photo of the crystal bulk. **d** High-resolution TEM image of the *ab*-plane confirming good single crystalline, with scale-up capture in the inset. **e** The simulated and **f** the measured electron diffraction image of monoclinic CCPS structure along the *c* axis, showing high consistency. The hexagonal patterns connected by dotted red lines in (**e**) and (**f**) correspond to the one connected by dotted black lines in (**b**).

from the highly mobile Cu ions-meditated migration, which could be manipulated by poling time, polarization, and current direction. The physical model of the ionic migration process in current rectification is illustrated in Supplementary Fig. 3. More important, the anisotropic conductivity are determined to come from the diffusion barrier in the crystal lattice, which affects the electrical potential[31] and causes different Cu ions migration efficiencies[32,33].

The schematic diagram of physical mechanism responsible for the in-plane electrical anisotropy is presented in Fig. 2g. The diffusion barrier is dependent on the energy level difference in specific transport direction, such as the *a* or the *b* axis. According to our electrical results, the diffusion barrier of the *a* axis ($\varphi_a$) is smaller than that of the *b* axis ($\varphi_b$), resulting in a more conductive channel along the *a* axis. There are two identical Schottky barriers $\varphi_1 = \varphi_2$ in the initial state, since the two electrodes of our device on both sides are Pt. Under the positive voltage poling, the electric field will drive Cu ions to conduct directional migration, resulting in the accumulation of Cu ions at the cathode interface to generate "Cu ions-rich" region. Thus, the Schottky barrier $\varphi_1$ is reduced by ionic capacitance effect, and causes $\varphi_1 < \varphi_2$. As shown in the bottom half of Fig. 2g, more Cu ions are accumulated when voltage is applied along the *a* axis compared to that along the *b* axis, thus $\varphi_1 < \varphi_3$ and $\varphi_2 > \varphi_4$ are indicated. The explored features not only prove the electrical anisotropy within the vdW layer of multiferroics, but also discover a superior candidate for artificial bionic synapses with multi-functional modes and multi-terminal directions[34–36].

The AFM vdW materials provide good opportunities for future memory, logic, and communication devices owing to their highly tunable magnetic properties. The switching of the Néel vector has stimulated the development of potential spintronic devices for multi-level neuromorphic outputs[37–39]. CCPS is reported to be an A-type AFM vdW multiferroic, in which Cr atom contributes to a long-range spin ordering, as its 3*d* shell is not fully filled, with 3 unpaired *d* electrons per atom[23,24]. When it is above Néel transition temperature ($T_N$), CCPS is in the paramagnetic phase, where the spins of Cr ions show a disordered state. As cooling down to below $T_N$, CCPS goes into the AFM phase, and the spins are oriented in opposite directions of alternating interlayers. The intralayer FM coupling and interlayer AFM coupling could induce a rich variety of GHz-frequency dynamical modes[40–42], which motivates us to investigate the uniaxial magnetic anisotropy and multiple spin orders. Magnetic measurements of were performed in the commercial superconducting quantum interference device (SQUID) and the physical property measurement system (PPMS). Temperature dependence of magnetic susceptibility ($\chi$ ~ *T*) in zero-field-cooling (ZFC) along the respective *a*, *b*, and *c* axes are shown in Fig. 3a, indicating the AFM interlayer coupling in CCPS. An evident cusp indicating $T_N = 32$ K is obtained in all directions, and a sharp drop of magnetization is observed at low temperature only along the *a* direction. It suggests that the spin orders prefer to lie along the *a* axis, which is the easy axis of magnetization. The *b* and *c* axes are hard axes, and their $\chi$–*T* curves overlap with much smaller magnetization change at low temperature. The dotted line is corresponding to the right axis $\chi$

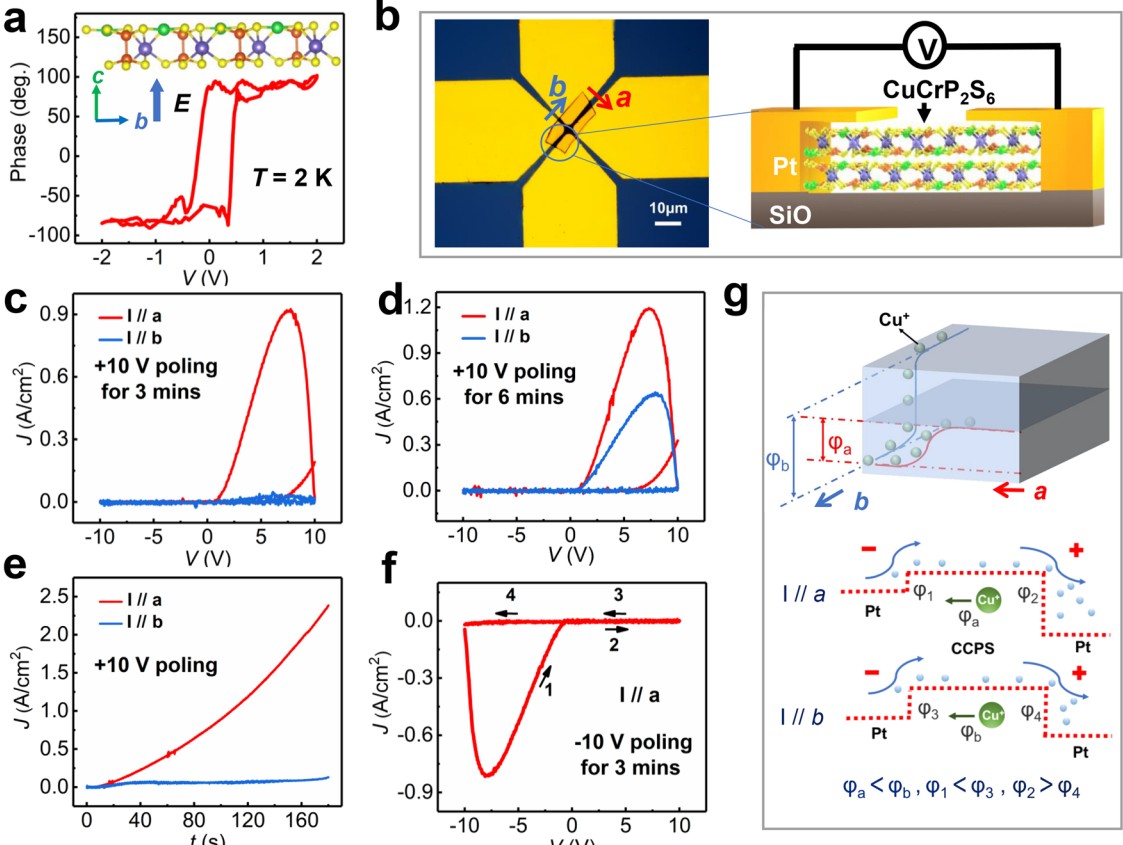

**Fig. 2 | Electrical measurements of the AFE CCPS indicating local FE polarization and electrical anisotropy. a** Phase *vs* voltage hysteresis loop of CCPS crystal bulk obtained at $T = 2$ K by PFM, after subtracting the built-in electric field between the sample and substrate. The upper inset diagram indicates the atomic structure of electric field ($E$) driven FE polarized state. **b** The microscope picture of our CCPS device with identified *a* and *b* axes, along with the scale-up configuration of electrical measurements on the right. **c** The rectifying *J–V* characteristics along the *a* and *b* axes as positive +10 V bias is poling for 3 min. **d** The rectification behaviors as increasing the poling voltage to 6 min or longer. **e** Time dependence of current density along the *a* and *b* axes as setting the poling bias +10 V. **f** The negative rectification along the *a* axis as the poling voltage is −10 V for 3 min. **g** The schematic diagram of physical mechanism illustrating the in-plane electrical anisotropy.

$^{-1}(H/M)$, showing the Curie-Weiss fitting. The Curie-Weiss temperature ($\theta_{CW}$) is estimated to be 24 K by $\chi = C/(T - \theta_{CW})$, wherein $C$ represents the Curie constant. The positive $\theta_{CW}$ demonstrates the FM intralayer coupling in this compound.

The hysteresis loops ($M–H$) along the *a*, *b*, and *c* axes at 10 K, 50 K, and 100 K are shown in Fig. 3b. They are obtained in a wide magnetic field range from −14 T to +14 T. At 10 K, the sample is saturated at a field of 7.4 T along the *c* axis, while the saturation field is 6 T in the *ab*-plane. It reveals the forced FM alignment by large field in AFM phase as $T < T_N$, weaker in-plane anisotropy and strong out-of-plane anisotropy. The *a* axis is the easy axis with minimum energy, and the anisotropic energy of the *b* axis is a little bit higher, while the *c* axis is the hard axis with significantly large energy. At 50 K and 100 K, the $M–H$ loops show high coincidence along the *a*, *b*, and *c* axes due to the paramagnetic phase as $T > T_N$. The scale-up $M–H$ minor loops along the three directions at 10 K are displayed in Fig. 3c. It is worth noting that there is an obvious in-plane spin reorientation transition marked in shadow, which represents a spin-flop (SF) transition. The SF transition presents an S-shaped curve and only happens in small field along the *a* axis as $T < T_N$. It reveals the 90° reorientation of the AFM Néel vector with a threshold field $H_{SF} \sim 0.4$ T, known as the SF field. The schematic diagram of the SF transition is presented in Fig. 3d, indicating the in-plane spin reorientation with anisotropic energy distribution and out-of-plane spin inverse parallel arrangement between vdW layers. The magnetic moment (spin order) is carried on the Cr site, denoted by the arrows. As $H_{SF}$ is achieved, the Néel vector rotates from the *a* axis to the *b* axis, lying perpendicular to the applied external field.

The magnetic easy axis is along the *a* direction, which should have the minimum anisotropic energy. The *b* axis has a slightly higher energy than that of the *a* axis, resulting in SF transition under a small field. The interlayer spin orders maintain AFM arrangement before and after the SF transition. This SF transition provides another evidence of higher-order magnetic anisotropy, which comes from the intrinsic special symmetry-breaking crystal structure of CCPS.

It is widely assumed that uniaxial magnetic anisotropy is critical for achieving ultrafast spin transfer and efficient spin-wave excitation in an antiferromagnet[40–43]. To investigate the magnon modes coming from the magnetic anisotropy and the SF transition, antiferromagnetic resonance (AFMR) measurements were proceeded at different temperatures and magnetic fields. We observed all the microwave absorption spectra in the frequency range of 3–13 GHz. The much lower frequency compared to traditional antiferromagnets is because of the dominant weak interlayer AFM coupling[43]. The representative resonances of 50 K and 10 K along the *a* axis are shown in Fig. 4a, b, respectively.

At 50 K ($T > T_N$) in Fig. 4a, there is a single peak in microwave absorption among all the frequencies, with gradual and linear shift from 0.1 T to 0.5 T as increasing the resonance frequency. The resonance signal at 100 K indicates similar behavior with that at 50 K (see Supplementary Fig. 5). In contrast, obvious double peaks are observed in the same frequencies at 10 K, as shown in Fig. 4b, which cannot be fitted by Kittel equations. The two resonance modes in dashed lines have the opposite changing trends, which come from different motion behaviors of spin orders, identified as different magnon modes.

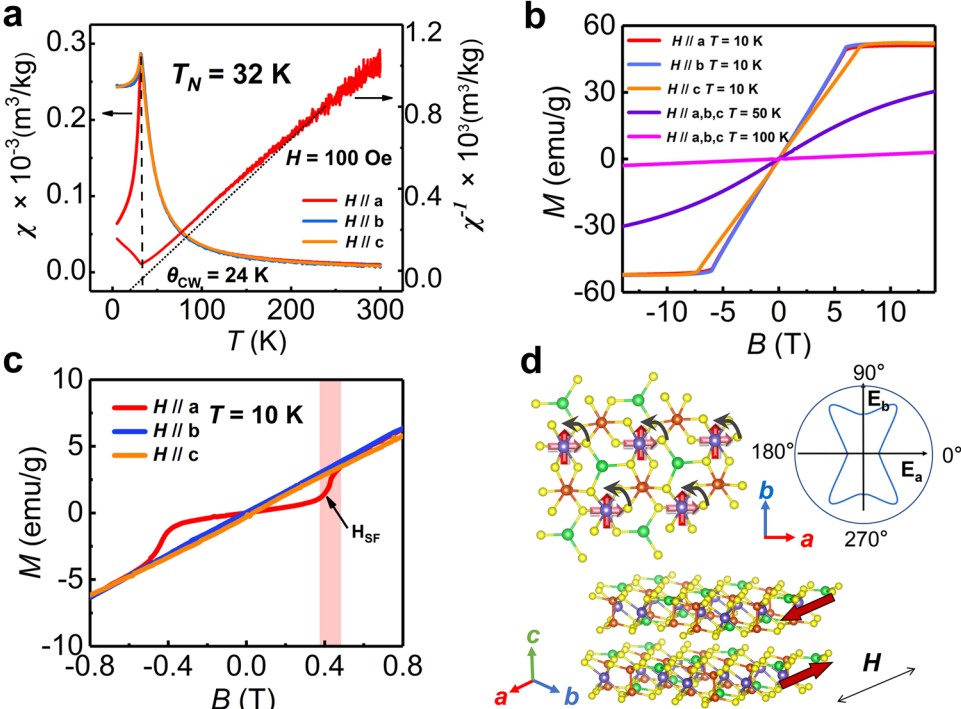

**Fig. 3 | Magnetic properties of CCPS indicating the magnetic anisotropy and the spin-flop (SF) transition. a** Temperature dependence of magnetic susceptibility ($\chi - T$) along the $a$, $b$, and $c$ axes. The dotted line corresponding to the right axis is Curie-Weiss fitting. **b** Magnetic field dependence of magnetization ($M - H$) curves along the $a$, $b$, and $c$ axes at different temperatures. **c** The scale-up minor loops along the three directions at 10 K, indicating the SF transition. **d** The schematic diagram of the SF transition indicating intralayer spin rotation, interlayer AFM orders and the distribution of anisotropic energy in the $ab$-plane.

For one mode, the magnetic field $H$ is decreasing as increasing the resonance frequency, while the other mode shows increasing field dependent on increasing frequency. The AFMR measurements along other directions, such as $b$ axis, do not show the similar multiple resonance modes, which are presented in Supplementary Fig. 6. The results confirm that the multiple magnon modes only exist along the $a$ axis as $T < T_N$.

We summarize the magnetic field dependence of the resonance frequency along the $a$ axis at 10 K, 50 K, and 100 K in Fig. 4c. It reveals the linear dependence at both 50 K and 100 K, indicating an electron paramagnetic resonance signal[44]. The shaded area between two specific frequencies of magnon modes at 10 K reaches a high degree of consistency with the SF transition range in shadow (0.38–0.48 T) of Fig. 3d. The Landau–Lifshitz model for A-type AFM vdW crystal lattices[43] has been carried out to fit the frequency $vs$ field data at 10 K, in order to determine the parameters of magnetic anisotropy in CCPS. As magnetic field is parallel to the $a$ axis within the vdW layer, the spin orders initially stay anti-parallel to each other and align along this easy axis. For this initial spin configuration, we obtain the resonance frequencies before the SF transition as

$$\omega_1 = \mu_0\gamma\sqrt{H^2 + H_b(H_E + H_c) + H_E H_c - \sqrt{H^2(H_b + H_c)(H_b + 4H_E + H_c) + H_E^2(H_b - H_c)^2}} \tag{1}$$

where $H_E$ is the interlayer exchange field, $H_b$ is the anisotropic field along the $b$ axis and $H_c$ is the anisotropic field along the $c$ axis. $H_a$ is set to be 0 as the reference point. As the external field is applied larger than $H_{RF}$, the resonance frequencies after SF transition are described as

$$\omega_2 = \mu_0\gamma\sqrt{\frac{\left(H^2\left(2H_E + H_b\right) - H_b\left(2H_E - H_b\right)^2\right)(2H_E + H_c - H_b)}{(2H_E - H_b)^2}} \tag{2}$$

Based on the Eqs. (1) and (2), we have fitted our experimental frequency $vs$ field data in Fig. 4c, and obtained the magnetic anisotropy parameters of $\mu_0 H_E = 5.334$ T, $\mu_0 H_b = 0.018$ T, and $\mu_0 H_c = 2.455$ T. The SF field is calculated as $\mu_0 H_{SF} = \sqrt{(2H_E - H_b)*H_b} = 0.433$ T, which is in good agreement with our observed $\mu_0 H_{SF}$ in the magnetization and AFMR measurements.

The schematic representation of low-energy magnon modes during the SF transition is illustrated in Fig. 4d. It displays different motion trajectories and spin orientations above and below the critical SF field ~$H_{SF}$. Despite the opposite alignments between two spin sublattices are in equilibrium, they undergo circular precessions in the same direction (left-handed or right-handed). Our density functional theory (DFT) calculations were performed using the projector augmented wave (PAW) as implemented in Vienna ab initio simulation package (VASP)[45,46]. The Perdew-Burke-Ernzerhof functional[47] was used for exchange and correlation potentials. We simulated the AFM resonance results by DFT calculations in Fig. 4e. The 7 exchange parameters ($J$) shown in Supplementary Table 1 are derived through fitting the total energy of 14 different spin structures in Supplementary Fig. 7. The final cross-validation score is less than 0.3 meV/f.u.[48], confirming the effectiveness of the obtained $J$ values. The extracted parameters demonstrate intralayer FM coupling and interlayer AFM coupling in CCPS, which are consistent with our experimental results. The magnon frequency at gamma is derived from the spinW package[49] by solving the spin Hamiltonian where all the exchange terms, anisotropy terms, and the magnetic field terms are taken into consideration. The underlying mechanism of multiple magnon modes during the SF transition is well explained.

In summary, the local in-plane electrical and magnetic anisotropies with the same easy axis in vdW CCPS have been explored in this work. Through systematical studies on this 2D multiferroic material, Cu ions-mediated polarization and migration are achieved, resulting in various rectifying characteristics depending on in-plane crystal

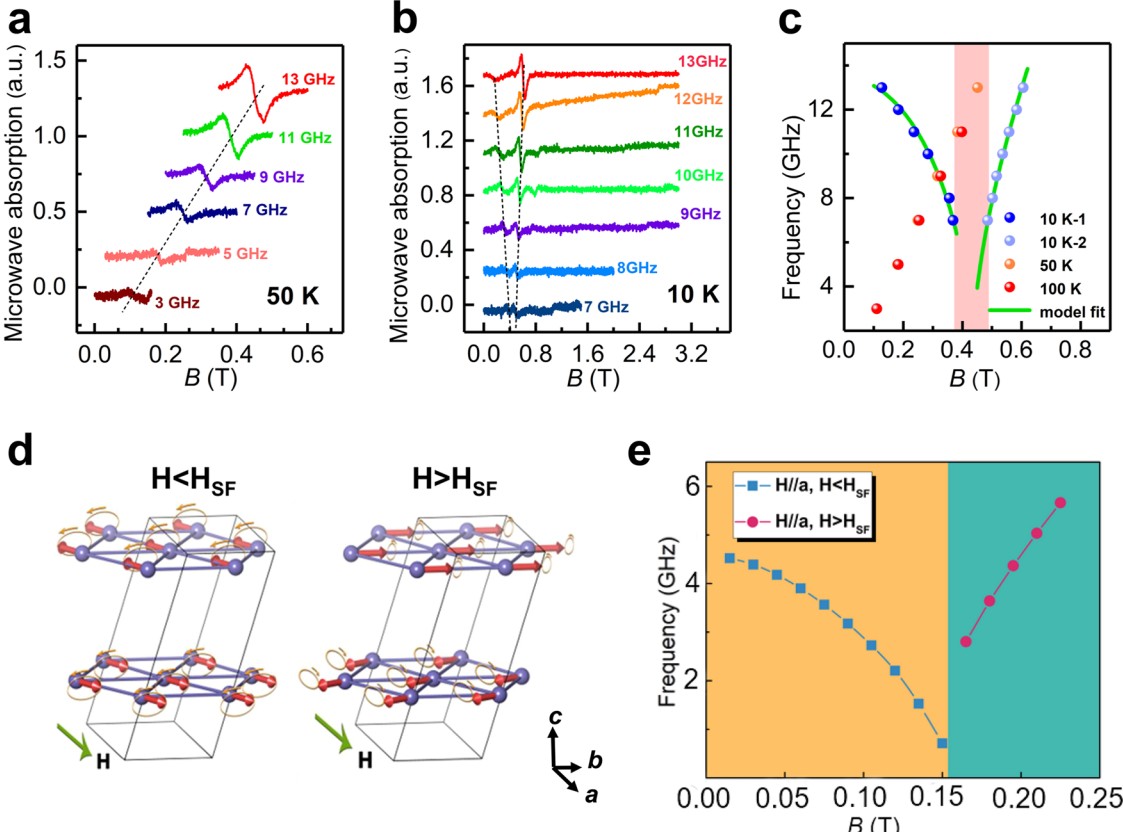

**Fig. 4 | The resonance modes associated with the SF transitions as $H \parallel a$.** The representative field dependence of absorption spectra with frequency 3–13 GHz at **a** 50 K and **b** 10 K by AFMR measurements. **c** The magnetic field dependence of resonance frequency at different temperatures, with fitted result at 10 K by Landau –Lifshitz model. The field interval between two special resonance modes at 10 K is marked as shadow, corresponding to the SF transition. **d** Schematic representation of different magnon modes below and above the SF field. **e** The calculated magnon frequency *vs* magnetic field by DFT.

directions and applied voltage polarities. The electrical anisotropy comes from different diffusion barriers along the distinct axes, which interact with the crystal lattice at the nanoscale in the layered flakes. Magnetic characterization and resonance measurements are provided to study the uniaxial magnetic anisotropy, which originates from the spin orientations of Cr sites. Moreover, the spin-flop transition of Néel vector accompanied with highly consistent resonance modes are intriguingly discovered and adequately analyzed. The model fittings and theoretical calculations are both conducted to sufficiently evidence the special magnon modes, and quantitatively provide the parameters of magnetic anisotropy. Our findings reveal the in-plane anisotropy of the vdW multiferroic CCPS, which are believed to play critical roles in the magnetoelectric coupling. This study will surely advance the development of emerging 2D multiferroic family, which provides potential applications for non-volatile storage devices with multi-functional modes, multi-terminal operations, multi-directional recognition, and multi-level outputs.

## Methods

### Crystal Preparation

High-quality single crystals of CCPS were synthesized by the chemical vapor transport method. The elements in the stoichiometric proportions of Cu:Cr:P:S = 1:1:2:6 were intimately mixed. The mixture was sealed in a vacuumed quartz tube, which was placed horizontally in a two-zone tube furnace. The temperature was set to 720 °C and 680 °C for the reaction and crystal growth region, respectively, for 8 days, followed by a furnace cooling to room temperature. The dark and flexible crystals with the size of 1.5 cm were harvested.

### Sample characterization

The crystallographic orientation of CCPS was obtained from an X-ray diffractometer (XRD, Bruker D8 Advance) using Cu Kα radiation. The high-quality crystalline and monoclinic structure of CCPS were confirmed by transmission electron microscope (TEM, JEM F200) operated at 200 kV. The samples were prepared by dropping the alcohol dispersion onto the carbon-coated copper TEM grids and atmospheric drying. Low-temperature piezoresponse force microscope (PFM) measurement was conducted using a commercial atomic force microscope (attoAFM I, attocube) employing a commercial electrostatic tip (Nanosensors, PtSi-FM) based on a closed-cycle helium cryostat (attoDRY2100, attocube).

### Device fabrication and electrical measurements

The thin flakes were achieved by mechanical exfoliation from synthetic bulk crystals onto SiO₂ (300 nm)/Si substrates transferred by PDMS (polydimethylsiloxane). The thickness of the flakes was identified by a Stylus Profiler (DektakXT, Bruker). By using photolithography and sputtering on positioned CCPS flake, the in-plane devices were fabricated with four Pt electrodes. Our device design resulted in a transport channel for homogeneous current flow, and the distance between the nearest electrodes was 2 μm. DC bias of 10 V was applied on the different adjacent electrode pairs to determine the current along *a* and *b* axes, resulting in various rectifying characteristics depending on different directions and voltage polarities.

### Magnetic characterization

Temperature dependence of magnetic susceptibility and hysteresis loops along the *a, b,* and *c* axes were collected at different

temperatures. All the measurements were performed in a super-conducting quantum interference device (SQUID, Quantum Design) equipped with a 5 T magnet and a physical property measurement system (PPMS, Quantum Design) with a 14 T magnet. We fixed the CCPS crystal on the sample holder, parallel to different crystal-lographic orientations.

## Antiferromagnetic resonance (AFMR) measurements

All the antiferromagnetic resonance (AFMR) measurements were proceeded in a PPMS-Dynacool system equipped with Phase FMR-40 at different temperatures and magnetic fields, by varying the microwave (MW) frequencies from 3 to 13 GHz. We investigated the spin resonance modes of CCPS by placing the crystal on a coplanar waveguide and measuring the microwave absorption spectrum by a two-port vector network analyzer. The radio frequency field was parallel to the external magnetic field during all the measurements, with the frequency of 1000 Hz and magnitude of approximately 0.45 Oe. The MW absorption signal was recorded when decreasing the external field.

## Density functional theory (DFT) calculation

The projector augmented wave (PAW) was carried out by using Vienna ab initio simulation package (VASP). The exchange and correlation potentials were described by Perdew-Burke-Ernzerhof functional. The kinetic energy cutoff for the plane-wave basis set is 400 eV and the energy threshold for electronic convergence is set to $10^{-6}$ eV. All atoms are fully relaxed until the Hellman-Feynman forces on each atom reach less than 0.01 eV/Å.

## Data availability

The data that support the findings of this study are available from the corresponding authors upon request.

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

## Acknowledgements

The authors acknowledge the experimental help by Prof. Genfu Chen in National Lab for Superconductivity at the Institute of Physics (Chinese Academy of Sciences) and Prof. Zhipeng Hou in South China Academy of Advanced Optoelectronics at South China Normal University. X.L.W. acknowledges the support by National Natural Science Foundation of China (12074018). H.L. acknowledges the supports by National Natural Science Foundation of China (62174044) and Guangdong Basic and Applied Basic Research Foundation (2022A1515010940). X.Y.W. acknowledges the support by National Key Research and Development Program of China (2019YFA0307900), National Natural Science Foundation of China (92163101) and Beijing Natural Science Foundation (Z190011). Y.-J.Z. acknowledges the supports by National Natural Science Foundation of China (52273298) and the Shenzhen Science and Technology Foundation (JCYJ20180507182246321, JCYJ20210324095611032 and JCYJ20220818100008016). J.G. and Z.C. acknowledge the supports by the Fundamental Research Funds for the Central Universities and the Research Funds of Renmin University of China (21XNLG27 and 22XNH097). H.D. acknowledges the supports by National Natural Science Foundation of China (61922077 and 11874347).

## Author contributions

X.L.W., X.Y.W., H.L., and Y.-J.Z. conceived and supervised the research work. X.L.W., Z.S., W.T., Y.-J.Z., S.T., B.W., and Y.X. developed the device fabrication and electrical measurements. C.Z. and H.D. performed DFT calculations. J.K., X.Y.W., and J.H. conducted the sample growth and XRD characterization. X.L.W., Z.S., G.W., J.D., and T.Z. proceeded the magnetic property measurements. T.L., S.C., and H.L. performed the FMR experiments. J.G. and Z.C. carried out the PFM measurement. L.L., D.P., and J.Z. provided the TEM characterizations. All authors discussed the results and commented on the manuscript.

## Competing interests

The authors declare no competing interests.
