## [Peer Review File · Nature Communications]

Electrical and Magnetic Anisotropies in van der Waals Multiferroic CuCrP2S6REVIEWER COMMENTS

Reviewer #1 (Remarks to the Author):

X. Wang and co-workers report the electrical and magnetic anisotropies of vdW multiferroic material CuCr₂S₆, which consists of alternating Cu and Cr layers. The Cu²⁺ ions are responsible for antiferroelectricity, while the Cr³⁺ ions bring about antiferromagnetic ordering at TN=32 K.

The key finding of the present work is identifying the in-plane uniaxial anisotropies of current rectifications and the bipolar rectification behavior at room temperature. The good electric conduction along the a axis seems to be a natural consequence that Cu migration is amenable to the Cu chains. Besides, the authors further employed several experimental techniques, including magnetic susceptibility, magnetization, and ESR, to elucidate magnetic anisotropy and magnon modes. In the previous works, the magnetic easy axis and the spin-flop transition are well established. The authors attempted to refine their behavior, yet it was not thoroughly pursued. Overall, I think the electrical anisotropy controlled by poling time, polarization, and current direction is an interesting scientific result. But the magnetic anisotropy part is a typical result. Therefore, I judge that this work does not meet the high standards of Nature Communications and cannot recommend it for publication. But the manuscript contains genuine scientific results, which could be more suitable for specialized journals. Below I provide reasoning for my judgments.

1. The authors should have mentioned that CuCr₂S₆ is primarily an antiferroelectric material throughout the main text. Ferroelectricity is merely a secondary effect. For readers, the complete description may need to be clarified.
2. Electrical and magnetic anisotropies with the same easy axis are merely coincidences, as they arise from two independent subsystems. This renders it impossible to control multiple degrees of freedom simultaneously. Actually, the authors treated the electrical and magnetic properties in a separate manner without demonstrating their mutual coupling.
3. I found a substantial difference between the experimental and calculated frequency-field diagrams shown in Figs. 4c and 4e. I think the magnetic parameters obtained from DFT should be fine-tuned to minimize this discrepancy. In addition, some additional analysis is needed to fit the frequency-field data in order to determine magnetic anisotropy. By the way, the authors mixed AFMR with FMR. Since CuCr₂S₆ is an antiferromagnetically ordered system, AFMR is the right term.

Reviewer #2 (Remarks to the Author):

The authors reported electrical and magnetic anisotropies in van der Waals multiferroic CuCr₂S₆, and investigated spin-flop transition corresponding to specific magnon modes. It exploits the superiority of multi-functionalities to manipulate the ferroelectric and ferromagnetic orders by electric direction/polarity, temperature variation, and magnetic field. In my opinion, the results presented in the manuscript are interesting to broader audiences and could bring new insight into solid devices. I therefore recommend that this work can be considered for publication in Nature Communications after minor revision.

1. The authors have studied electrical anisotropy by PFM measurement and rectification characteristics. Is the ferroelectric "Phase vs V" loop related to rectifying "J vs V" loop in Fig.2? It should be clarified.

2. The authors claim that the electrical anisotropy originates from Cu^+ ions migration, and provide the physical model of diffusion barrier along a and b axes in Fig. 2g. But it lacks sufficient evidence to support this mechanism.

3. In Fig. 3c, the saturated field along c axis is larger than that along a and b axes. Why is it hard to force FM alignment along c axis? Why the MH curves of a and b coincide? The authors should provide more explanations in the manuscript.

4. What are the spin orientations above and below HSF? The authors should specify the actual directions and mark the crystal axes in Fig. 4d.

5. The FMR results are interesting. The authors should provide more information, such as the magnetic damping coefficient and anisotropic constant for this novel multiferroic material.

Reviewer #3 (Remarks to the Author):

The authors studied a new van der Waals material CuCrP_2S_6 , which has recently attracted lots of attention due to the coexistence of ferroelectric and magnetic properties. Understanding of the physical properties in this kind of material is both important for fundamental studies of electrical and magnetic polarisations, and for practical application as the authors suggested in the paper. The paper reported new electrical and magnetic studies, as well as magnetotransport studies in nanodevices. Their findings are promising, and in my opinion, will attract the interest of NC readers and lead to further interesting exploration along the line. However, there are questions that I think should be clarified.

1. My understanding of the result is the following:

It showed there is anisotropy in electrical transport, which happens in the same plane as the magnetic component. ie there is relation between electrical conductivity and magnetic property. Could the author say something about the effect of ferroelectric polarisation on magnetic property and vice versa.

2. Line 130 and Fig. 1f: which little twist is it referring to?

3. Fig. 1a: Is the crystal structure correct? Because some of the S atoms only bonds to one side, but not the other side. It seems there is missing chemical bonds. Also are the position of the Cu ions correct? When in the ferroelectric phase, shouldn't all the Cu ions be either on the top layer or the bottom layer like in CuInP_2S_6 ?

4. Line 145: Is this a crystal or flake? Do the authors mean polarisation along the c direction ie, out-of-plane rather than in-plane (ab plane)?

5. Fig 2a: Does figure (a) an out of plane dipole, and the rest of the figures are in plane?

6. Fig 2c,d: a axis is shorter than b axis, how does it compare if the applied voltage is normalized against their lengths?

7. Line 191: Am I understanding it right, that the applied voltage only changes the barrier height at the contact/vdw interface, but not the in-plane polarisation of the vdw layer.

8. Line 222: shouldn't M be aligned along a axis at low temperature if a is the easy axis? if yes, should M increases at low T rather than a drop?

Point-by-point Response to Reviewers

Reviewer #1 (Remarks to the Author):

X. Wang and co-workers report the electrical and magnetic anisotropies of vdW multiferroic material CuCr_2S_6 , which consists of alternating Cu and Cr layers. The Cu^{2+} ions are responsible for antiferroelectricity, while the Cr^{3+} ions bring about antiferromagnetic ordering at $T_N=32$ K.

The key finding of the present work is identifying the in-plane uniaxial anisotropies of current rectifications and the bipolar rectification behavior at room temperature. The good electric conduction along the a axis seems to be a natural consequence that Cu migration is amenable to the Cu chains. Besides, the authors further employed several experimental techniques, including magnetic susceptibility, magnetization, and ESR, to elucidate magnetic anisotropy and magnon modes. In the previous works, the magnetic easy axis and the spin-flop transition are well established. The authors attempted to refine their behavior, yet it was not thoroughly pursued. Overall, I think the electrical anisotropy controlled by poling time, polarization, and current direction is an interesting scientific result. But the magnetic anisotropy part is a typical result. Therefore, I judge that this work does not meet the high standards of Nature Communications and cannot recommend it for publication. But the manuscript contains genuine scientific results, which could be more suitable for specialized journals. Below I provide reasoning for my judgments.

Reply: We thank the reviewer for the positive affirmation of our results to be interesting and genuine scientific, especially the original observation and modulation of electrical anisotropy. For the magnetic anisotropy part, we are sorry for not highlighting our innovation due to our imperfect description. Although there is one previous work reporting the magnetic easy axis and the spin-flop transition, cited as Ref. [23] in our manuscript, we provide in-depth investigation and quantitatively analyze the magnetic anisotropy. In particular, we discover corresponding magnon modes for applicable antiferromagnetic (AFM) spin torque nano-oscillators.

In the magnetic part, there are two highlights we have addressed originally and one more highlight we added to the revised manuscript: (1) We have conducted systematic antiferromagnetic resonance (AFMR) measurements in the van der Waals (vdW) multiferroic CuCrP_2S_6 for the first time, which has in-plane magnetic anisotropy with regard to the magnetic field direction relative to the crystal axes. The magnon resonances, representing AFM dynamics with well-defined easy-plane order are in an accessible GHz-frequency range which can be tuned by adjusting an in-plane magnetic field. (2) Through plotting the magnetic field dependence of resonance frequencies, we have identified the different regimes of magnon modes, corresponding perfectly to spin-flop transition in the magnetization (M-H) curve along the easy axis. Our DFT

simulations have provided a similar mapping of magnon frequency *vs* magnetic field, together with exchange parameters of intralayer FM coupling and interlayer AFM coupling. (3) As the reviewer's suggestion in comment 3, the Landau–Lifshitz model has been carried out to fit our frequency *vs* field data to quantify the parameters of magnetic anisotropy and determine the AFM dynamics. It would provide a good reference for future investigation in manipulating in-plane magnetic anisotropy of AFM vdW multiferroic materials and the fundamental understanding to utilize these features in future spintronic devices.

We have made a major revision of our manuscript with clear emphasis and thoroughly analysis on magnetic anisotropy. Moreover, we take a step further to provide the related parameters of the anisotropic field along different axes. We have improved Fig. 3 to clarify our focus on the discussion of magnetic anisotropy, and also added an additional model to fit our frequency *vs* field result in Fig. 4c. The specific characteristics of magnetic anisotropy are well described in this work for the first time. Hopefully the reviewer could consider the significance of this work again, which will be of broad interests to the diverse readership of Nature Communications. We appreciate all the concerned issues from the reviewer, which help to improve the manuscript a lot. The point-by-point response to his/her thoughtful comments are addressed in the following.

1. The authors should have mentioned that CuCr₂S₆ is primarily an antiferroelectric material throughout the main text. Ferroelectricity is merely a secondary effect. For readers, the complete description may need to be clarified.

Reply: Thanks for this important correction and sorry for our inaccurate expression in the main text. As the reviewer's precise description, CuCrP₂S₆ (CCPS) is primarily an antiferroelectric material, while its ferroelectricity is a secondary effect achieved by applying an electric field. We have revised our manuscript throughout according to the reviewer's suggestion, and clarified the complete description in a more scientific and accurate way as the following.

“CuCrP₂S₆ is known to be in the antiferroelectric (AFE) phase below the critical transition temperature $T_C \sim 145$ K [17,23,26]. The Cu ions occupy the upper and lower position alternatively, resulting in absence of spontaneous macroscopic polarization. The side-view structure (interlayer *bc*-plane) in the AFE state is indicated in Fig. 1a, while the the top-view structure (intralayer *ab*-plane) is shown in Fig. 1b.”

“The piezoresponse force microscope (PFM) measurement was conducted on a CCPS crystal bulk (thick flake of ~ 3 μm thickness). The saturated and symmetric hysteresis loop of phase *vs* DC bias voltage at low temperature was obtained, as shown in Fig. 2a. We observed the distinct 180° switching of ferroelectric polarization as the electric field is applied out-of-plane (along the *c*-axis). The structure diagram of emerged metastable ferroelectricity (FE) is represented in the upper inset of Fig. 2a,

consisting of Cu atoms lying on the same side in a vdW layer. The out-of-plane electric dipole could originate from the AFE domain boundaries with unbalanced charges, which are related to defect-dipole polarization under the application of an electric field [12, 24, 25]. ”

To be more clear for readers, we have improved Fig. 1a to indicate the Cu ions sitting on the upper and lower positions alternatively in the typical AFE state, and added an inset figure in Fig. 2a to illustrate the electric field driven FE state with Cu atoms lying only on the upper side.

2. Electrical and magnetic anisotropies with the same easy axis are merely coincidences, as they arise from two independent subsystems. This renders it impossible to control multiple degrees of freedom simultaneously. Actually, the authors treated the electrical and magnetic properties in a separate manner without demonstrating their mutual coupling.

Reply: We thank the reviewer for bringing this important comment to our attention. We agree that electrical and magnetic anisotropies with the same easy axis might be coincidences. They don't seem to be related because the electrical anisotropy originates from Cu atoms, while magnetic anisotropy originates from Cr atoms. However, there are a few pieces of evidence to support their mutual coupling.

(1) From a theoretical point of view, there is a real opportunity to achieve magnetoelectric coupling in CCPS system by considering crystal structure symmetry. Whether electric field driven Cu⁺ polarization, or the diffusion barrier-induced anisotropic Cu⁺ migration, the symmetry of the crystal is proved to be broken, resulting in Rashba-spin-orbit coupling (Rashba-SOC). On the other side, magnetic anisotropy comes from SOC, so electrical anisotropy should be coupled with magnetic anisotropy. Through electric polarization, the Rashba-SOC produced Hamiltonian by symmetry breaking can be written as

$$H_{\text{SOC}} \propto a(k_y \sigma_x - k_x \sigma_y)$$

The in-plane equivalent field of Rashba-SOC generated by electric polarization could change the magnetic anisotropic energy, resulting in magnetic and electric mutual coupling.

(2) This material is known to be a type-II multiferroic material, with strong polarization-magnetization coupling according to Ref. [17-24] in the manuscript. The ferroelectricity stems from the inversion symmetry breaking driven by spin order. More experimental and theoretical evidence was provided by several research groups recently in Ref. [20, 23, 26] of our text, in which the electric dipoles and spins are bridged by spin-orbit coupling. Magnetic field induced electric polarization and electric field manipulated magnetic orderings should exist even in a single layer limit of CCPS.

(3) In our experiment, the magnetoelectric coupling in our CCPS bulk crystal was

demonstrated by magnetoelectric current *vs* magnetic field measurement, as shown in Supplementary Fig. 1. The I_{ME} *vs* $\mu_0 H_{\text{ab}}$ curve exhibits two peaks at the positive and negative fields, which indicate the magnetoelectric coupling in bulk CCPS. It should be noted the I_{ME} is only around 100 pA range for the bulk CCPS (typically for single-phase multiferroics). We did try to measure I_{ME} on a thin flake device in different directions. Unfortunately, it seems that the weak I_{ME} on a nanoflake cannot be probed by the current PPMS technique, nor can we study whether the easy axis has any impact on the magnetoelectric coupling.

In this work, for the first time, we demonstrate the electrical anisotropy in multiferroic CCPS. We also observe the spin flop (SF) transition of the Néel vector along the *a* axis accompanied by highly consistent magnon modes which are firstly discovered and adequately analyzed. These findings are important to this emerging 2D vdW multiferroic family. We believe that these anisotropies play critical roles in the magnetoelectric coupling, which requires further investigations based on more techniques, such as spin-polarized scanning tunneling microscopy/spectroscopy. Furthermore, we also plan to design experiments to prove the coupling and mutual regulation between magnetic and electrical anisotropies, which are electronically manipulated magnetic anisotropy and magnetically modulated electrical anisotropy respectively. These are important research topics in multiferroics with the opportunity to realize practical control of multiple degrees of freedom simultaneously.

We again thank the reviewer for raising this essential point and providing multiple perspectives on this issue, which is worth to be investigated and figured out in future.

3. I found a substantial difference between the experimental and calculated frequency-field diagrams shown in Figs. 4c and 4e. I think the magnetic parameters obtained from DFT should be fine-tuned to minimize this discrepancy. In addition, some additional analysis is needed to fit the frequency-field data in order to determine magnetic anisotropy. By the way, the authors mixed AFMR with FMR. Since CuCr2S6 is an antiferromagnetically ordered system, AFMR is the right term.

Reply: We appreciate the insightful and constructive comments from the reviewer, which help to improve our manuscript a lot.

Firstly, our DFT calculation was directly based on the fixed material structure. The exchange correlation *J* parameters were fitted among Cr sites through different spin configurations. The anisotropic tensor was obtained by considering the spin-orbit coupling (SOC) effect, and it was taken into spin Hamiltonian to solve the spin wave frequency at the Γ point. The first-principle calculations are dependent on actual materials and do not rely on superfluous empirical parameters, resulting in intuitive microscopic parameters. The spin wave frequencies calculated by DFT do have certain deviations from our experimental results, which could come from the following reasons. (1) The real material is under complex surroundings in experiments. The DFT

calculations only consider a relatively simple environment of the ground state, in which many influencing factors are ignored, such as temperature, defect, interface effects, etc. Thus, it is actually a big challenge to perfectly fit the experimental data by calculating from scratch. (2) The obtained spin Hamiltonian also requires many parameters, including 7 exchange parameters and 2 anisotropic parameters in our case. Such a multi-parameter problem itself could have a relatively large range of errors.

Nevertheless, the V-shaped curve of the experimentally measured resonance frequency ν vs applied magnetic field is fundamentally derived from magnetic anisotropy, which is well reflected by a coincided V-shaped curve in our DFT calculations. It demonstrates that the magnon resonance modes fitted by DFT calculations already have the same trend as the observation in the experiment, although the numerical discrepancy exists, which is difficult to avoid.

As the reviewer suggested, we have added an additional model and taken numerical analysis to fit our frequency ν vs field data to determine the parameters of magnetic anisotropy in CCPS. The Landau–Lifshitz model fitting for A-type AFM vdW crystal lattices is carried out according to Ref. [43] in the revised manuscript. As magnetic field is parallel to the a axis within the vdW layer, the spin orders initially stay anti-parallel to each other and align along this easy axis. For this initial spin configuration, we obtain the resonance frequencies before the SF transition as

$$\omega_1 = \mu_0 \gamma \sqrt{H^2 + H_b(H_E + H_c) + H_E H_c - \sqrt{H^2(H_b + H_c)(H_b + 4H_E + H_c) + H_E^2(H_b - H_c)^2}} \quad (1)$$

where H_E is the interlayer exchange field, H_b is the anisotropic field along the b axis and H_c is the anisotropic field along the c axis. H_a is set to be 0 as the reference point. As H_{RF} is achieved, the Néel vector rotates from the a axis to the b axis, lying perpendicular to the applied external field. The resonance frequencies after SF transition are described as

$$\omega_2 = \mu_0 \gamma \sqrt{\frac{(H^2(2H_E + H_b) - H_b(2H_E - H_b)^2)(2H_E + H_c - H_b)}{(2H_E - H_b)^2}} \quad (2)$$

Based on the equation (1) and (2), we have fitted our experimental frequency ν vs field data in Fig. 4c, and obtained the magnetic anisotropy parameters of $\mu_0 H_E = 5.334$ T, $\mu_0 H_b = 0.018$ T and $\mu_0 H_c = 2.455$ T. The SF field is calculated as $\mu_0 H_{SF} = \sqrt{(2H_E - H_b) * H_b} = 0.433$ T, which is in good agreement with the observed $\mu_0 H_{SF} \sim 0.4$ T in our magnetization and AFMR measurements. We have added our model fitting results to Fig. 4c in the revised manuscript, as shown in the following figure.

We have added the above Landau–Lifshitz model fitting to Fig. 4c and provided the reasonable magnetic anisotropy parameters in our revised manuscript. The in-plane magnetic anisotropy is weak. The small difference of anisotropic fields between the a and b axes is obtained, quantitatively demonstrating the mechanism of the SF transition. In contrast, the out-of-plane anisotropy is strong, which needs a large field to force spin alignment along the c axis. The relative magnitude of the above model fitted parameters is consistent with the relative magnitude of the DFT calculated values in Supplementary Table 1, which proves the rationality of both calculation methods. In this work, the thorough analysis and specific numerical parameters of magnetic anisotropy provide a good reference for future investigation in CCPS and a foundational understanding to utilize the feature in future AFM spintronic devices.

At last, sorry for the mixture of AFMR with FMR, we have revised it throughout the manuscript. Thanks very much for this correction from the reviewer.

Reviewer #2 (Remarks to the Author):

The authors reported electrical and magnetic anisotropies in van der Waals multiferroic

CuCrP₂S₆, and investigated spin-flop transition corresponding to specific magnon modes. It exploits the superiority of multi-functionalities to manipulate the ferroelectric and ferromagnetic orders by electric direction/polarity, temperature variation, and magnetic field. In my opinion, the results presented in the manuscript are interesting to broader audiences and could bring new insight into solid devices. I therefore recommend that this work can be considered for publication in Nature Communications after minor revision.

Reply: We appreciate the high evaluation and recommendation by the reviewer. We have addressed all the thoughtful comments as below.

1. The authors have studied electrical anisotropy by PFM measurement and rectification characteristics. Is the ferroelectric “Phase vs V” loop related to rectifying “J vs V” loop in Fig.2? It should be clarified.

Reply: We thank the reviewer for this comment. As the reviewer may concern, there are certain relationships between ferroelectric polarization and ionic migration in traditional ferroelectrics. In contrast, metal thiophosphates, such as CuInP₂S₆ and CuCrP₂S₆, have different characteristics owing to their layered structures, intermediate bandgaps, multiple ferroic orders and outstanding ionic conductivity. It is complicated to provide the direct intercorrelation between ionic kinetics and ferroelectric switching properties. In this work, we have realized local ferroelectricity in antiferroelectric CCPS by PFM measurement in Fig. 2a, indicating the out-of-plane electric dipole driven by applied electric field. The rectifying behaviors in Fig. 2c, 2d and 2f are demonstrated to originate from the highly mobile Cu⁺ mediated migration, which could be manipulated by poling time, polarization and current direction. Thus, the ferroelectric polarization is not the only factor that causes ion migration. But for both Cu⁺ polarization and Cu⁺ migration, the symmetry of the crystal is proved to be broken, resulting in electrical anisotropy. This is the common feature of “Phase vs V” curve and “J vs V” loop. Furthermore, the interplay coupling and mutual manipulation of ferroelectricity and ionic conductivity in CCPS may stimulate new ideas and concepts in 2D vdW ferroelectric and multiferroic devices.

2. The authors claim that the electrical anisotropy originates from Cu⁺ ions migration, and provide the physical model of diffusion barrier along a and b axes in Fig. 2g. But it lacks sufficient evidence to support this mechanism.

Reply: We thank the reviewer for this comment. We did try to calculate the diffusion barriers along the *a* and *b* axes to confirm the electrical anisotropy in the *ab*-plane by the climbing-image nudged elastic band (NEB) method as implemented within Vienna ab initio simulation package (VASP). However, we didn't reach consistent outcomes with our experimental data. It is because that NEB considers the migration path and diffusion barrier between two stable states, and cannot calculate a reasonable value in polarized state.

Moreover, the in-plane diffusion pathway of Cu ions migration was recently reported [1] by first-principle calculations, which displays the calculated activation barriers for in-plane diffusion of Cu ions in CCPS are larger than that of the lithium-ion conductor $\text{Li}_4\text{P}_2\text{S}_6$. Also, another work has performed local current measurements to reveal different diffusion paths for Cu ion migration of in-plane intralayer and out-of-plane interlayer [2]. Therefore, the different diffusion barriers induced in-plane electrical anisotropy is a natural consequence, as indicated by our provided physical model. We have added the related references in the following to our revised manuscript.

[1] Xu, D.-D. *et al.* Unconventional out-of-plane domain inversion via in-plane ionic migration in a van der Waals ferroelectric. *J. Mater. Chem. C* **8**, 6966-6971, (2020).

[2] Zhang, D. *et al.* Anisotropic ion migration and electronic conduction in van der Waals ferroelectric CuInP_2S_6 . *Nano Lett.* **21**, 995-1002, (2021).

3. In Fig. 3c, the saturated field along c axis is larger than that along a and b axes. Why it is hard to force FM alignment along c axis? Why the MH curves of a and b coincide? The authors should provide more explanations in the manuscript.

Reply: We thank the reviewer for this comment. The magnetic easy axis in CCPS is the a axis, and the secondary easy axis is the b axis, while the hard axis is the c axis. The smaller saturated field indicates weaker anisotropy in the ab -plane, that is why a spin flop (SF) transition could happen under a small field. In contrast, the large saturated field demonstrates that the c axis is the hard axis, with strong out-of-plane anisotropy. That is why it is hard to force FM alignment along this direction.

The difference in anisotropic field / energy between the a and b axes is very small, so it seems to coincide in the M-H curves under a 14 T magnetic range. The weak anisotropy in the ab -plane is also proved by the observation of SF transition, as the AFM Néel vectors easily switch the direction from the a axis to the b axis under a minor 0.4 T field. The a axis is the easy axis with minimum energy, the anisotropic energy of the b axis is a little bit higher, and the c axis is the hard axis with significantly large energy. As the reviewer suggested, we have added more explanations in the revised manuscript.

4. What are the spin orientations above and below H_{SF}? The authors should specify the actual directions and mark the crystal axes in Fig. 4d.

Reply: We thank the reviewer for this important reminder. We have added Fig. 3d and the following description to our revised version, in order to specify the spin orientations above and below H_{SF} more clearly.

“The schematic diagram of the SF transition is presented in Fig. 3d, indicating the in-plane spin reorientation with anisotropic energy distribution and out-of-plane spin inverse parallel arrangement between vdW layers. The magnetic moment (spin order) is carried on the Cr site, denoted by the arrows. As H_{SF} is achieved, the Néel vector rotates from the a axis to the b axis, lying perpendicular to the applied external field.

The magnetic easy axis is along the a direction, which should have the minimum anisotropic energy. The b axis has a slightly higher energy than that of the a axis, resulting in SF transition under a small field. The interlayer spin orders maintain AFM arrangement before and after the SF transition.”

Moreover, we have marked the crystal axes in Fig. 4d for clarity and consistency. Thanks again for this reminder.

5. The FMR results are interesting. The authors should provide more information, such as the magnetic damping coefficient and anisotropic constant for this novel multiferroic material.

Reply: We thank the reviewer for the positive affirmation and constructive suggestion about our FMR results. We have added an additional model and taken numerical analysis to fit our frequency *vs* field data in our revised manuscript. The Landau-Lifshitz model fitting to Fig. 4c for A-type AFM vdW crystal lattices is carried out according to Ref. [43]. We have calculated the anisotropic constants, such as interlayer exchange field and anisotropic fields along different axes, which quantitatively determine the magnetic anisotropy and reveal the physical mechanism of the SF transition in CCPS.

As we have mentioned in the manuscript, the observed special resonance modes cannot be fitted by Kittel equations. So it is not easy to obtain the damping coefficient of CCPS. To be more accurate, we change ferromagnetic resonance (FMR) to antiferromagnetic resonance (AFMR) in our revised manuscript.

Reviewer #3 (Remarks to the Author):

The authors studied a new van der Waals material CuCrP₂S₆, which has recently attracted lots of attention due to the coexistence of ferroelectric and magnetic properties. Understanding of the physical properties in this kind of material is both important for fundamental studies of electrical and magnetic polarisations, and for

practical application as the authors suggested in the paper. The paper reported new electrical and magnetic studies, as well as magnetotransport studies in nanodevices. Their findings are promising, and in my opinion, will attract the interest of NC readers and lead to further interesting exploration along the line. However, there are questions that I think should be clarified.

Reply: We thank the reviewer for the positive review and thoughtful comments which we address below.

1. My understanding of the result is the following:

It showed there is anisotropy in electrical transport, which happens in the same plane as the magnetic component. ie there is relation between electrical conductivity and magnetic property. Could the author say something about the effect of ferroelectric polarisation on magnetic property and vice versa.

Reply: We thank the reviewer for this good question. The magneto-electric coupling in our CCPS was demonstrated by magnetoelectric current vs magnetic field measurement, as shown in Supplementary Fig. 1. It should have strong correlation and interaction between electrical anisotropy and magnetic anisotropy in principle. Both electric dipoles and spin orders stem from spin-orbit coupling, originating from crystal symmetry breaking. We believe that the observed anisotropies play critical roles in the magnetoelectric coupling, which requires further investigations. Furthermore, we also plan to design experiments to prove the coupling and mutual regulation between magnetic and electrical anisotropies, which are electronically manipulated magnetic anisotropy and magnetically modulated electrical anisotropy respectively. We thank the reviewer for raising this comment and bringing new perspectives on this issue.

2. Line 130 and Fig. 1f: which little twist is it referring to?

Reply: Sorry for the confusion to the reviewer. The little twist is referring to the connected hexagon in Fig. 1f, which is not as regular as the simulated one in Fig. 1e. We have improved the description to be more clear in the revised manuscript.

3. Fig. 1a: Is the crystal structure correct? Because some of the S atoms only bonds to one side, but not the other side. It seems there is missing chemical bonds. Also are the position of the Cu ions correct? When in the ferroelectric phase, shouldn't all the Cu ions be either on the top layer or the bottom layer like in CuInP_2S_6 ?

Reply: Thanks for this important reminder from the reviewer. Sorry for our misleading structure of CCPS in Fig. 1a, which shields some chemical bonds around S atoms. We have replaced it with a clearer schematic diagram of atomic structure in the revised manuscript. For the position of Cu ions, we showed the antiferroelectric arrangement, where the Cu ions occupy the upper and lower position alternatively, resulting in absence of spontaneous macroscopic polarization. It is because that CCPS is primarily an antiferroelectric material, while its ferroelectricity is a secondary effect achieved by applying an electric field. It is different from the typical ferroelectric material CuInP_2S_6 .

To be clearer for readers, we have added an inset figure in Fig. 2a to illustrate the electric field driven ferroelectric state with Cu atoms lying only on the upper side.

4. Line 145: Is this a crystal or flake? Do the authors mean polarisation along the *c* direction ie, out-of-plane rather than in-plane (*ab* plane)?

Reply: We thank the reviewer for the correction. The sample for PFM measurement is a CCPS crystal bulk, which is actually a thick flake with $\sim 3 \mu\text{m}$ thickness. We observed the distinct 180° switching of ferroelectric polarization as the electric field is applied out-of-plane, which is along the *c*-axis instead of the *ab*-plane. Sorry for the writing typo and thanks again for the correction.

5. Fig 2a: Does figure (a) an out of plane dipole, and the rest of the figures are in plane?

Reply: We thank the reviewer for the question. The answer is yes. We used Fig. 2a to indicate the saturated and symmetric hysteresis loop of phase vs DC bias voltage at low temperatures, demonstrating the electric field driven ferroelectricity. As everyone knows, it is not easy to measure ferroelectric polarization in plane at low temperature by PFM. For the rest of the figures in Fig. 2, we show the anisotropic rectification behaviors in the *ab*-plane for the first time, while the out-of-plane anisotropic electronic conduction in a similar system CuInP_2S_6 has been reported in Ref. [33] in our revised manuscript.

6. Fig 2c,d: *a* axis is shorter than *b* axis, how does it compare if the applied voltage is normalized against their lengths?

Reply: We thank the reviewer for this comment. Although there are different lengths along the *a* and *b* axes, the channel is the same. The rectifying behavior only happens in the channel and depends on the conductive film part. That is why we normalized J (A/cm^2) to the size of the channel. Moreover, the difference proportion of electrical anisotropy reaches a ratio of ~ 18 as $+10 \text{ V}$ is poling for 3 min. So it should not depend on the difference proportion of length along the *a* and *b* axes.

7. Line 191: Am I understanding it right, that the applied voltage only changes the barrier height at the contact/vdw interface, but not the in-plane polarisation of the vdw layer.

Reply: We thank the reviewer for this comment. We think that the bipolar rectification in Fig.2f is caused by the in-plane polarization flipping, resulting in different barriers on both sides when the positive voltage is applied in different directions. So the applied voltage changes the in-plane polarization of the vdW layer, and the barrier height at the contact/vdW interface.

8. Line 222: shouldn't *M* be aligned along *a* axis at low temperature if *a* is the easy axis? if yes, should *M* increases at low *T* rather than a drop?

Reply: We thank the reviewer for this thoughtful comment. The drop is coming from the interlayer antiferromagnetic coupling. From the M-H loop around zero-field in Fig. 3c, one can see the slope along the a axis is smaller than the b and c axes. As $\chi = M/H$, where χ is the slope of M-H at zero field, it is obvious that χ of the a axis drops faster than the b and c axes. To be more accurate, it is the Néel vector which prefers to lie along the a axis at low temperatures. But after applying magnetic field as large as H_{SF} , it is easy to switch the a axis to the b axis due to the weak magnetic anisotropy and the SF mechanism in plane.

REVIEWERS' COMMENTS

Reviewer #1 (Remarks to the Author):

The authors have addressed most of my initial concerns and improved the description and analysis of the data. Further, the authors highlight the significant role of in-plane anisotropy in inducing magnetoelastic coupling. In the revised version, however, I expected more rigorous experimental verification of the mutual control of magnetic and electrical anisotropies. If this coupling is not proven, I cannot recommend the publication of this manuscript in Nature Communications.

The rationale for this decision:

First, it is a pretty standard exercise to analyze AFMR in AFM systems with uniaxial anisotropy, although the authors claim to have, for the first time, measured AFMR in the vdW multiferroic.

Second, electric and magnetic anisotropy is characterized in a separate temperature window. The sum of individual characterizations is insufficient to make a strong case in multiferroic materials.

Reviewer #2 (Remarks to the Author):

The authors have addressed all the concerns and revised the manuscript accordingly. Now I would recommend it for publication in Nature Communications.

Reviewer #3 (Remarks to the Author):

The authors have addressed my questions in the response and I find this work suitable to be considered for the publication in Nature Communications.

Point-by-point Response to Reviewers' Comments

Reviewer #1 (Remarks to the Author):

The authors have addressed most of my initial concerns and improved the description and analysis of the data. Further, the authors highlight the significant role of in-plane anisotropy in inducing magnetoelastic coupling. In the revised version, however, I expected more rigorous experimental verification of the mutual control of magnetic and electrical anisotropies. If this coupling is not proven, I cannot recommend the publication of this manuscript in Nature Communications.

Reply: We appreciate the positive confirmation of our previous responses and the new constructive comments from the reviewer. In this work, the innovation is the first discovery and multiple manipulation of in-plane electrical and magnetic anisotropies in van der Waals (vdW) multiferroic CuCrP_2S_6 (CCPS). We are not focusing on the magnetoelectric coupling, although it is another important effect for future investigation. Anyhow, we have demonstrated the magnetoelectric coupling in CCPS in Supplementary Fig. 1 through magnetoelectric current vs magnetic field measurement. The coupling of the electrical and magnetic anisotropies is foreseen but only from a theoretical point, as we replied in our last responses. However, the experimental demonstration of mutual control of the electrical and magnetic anisotropies is challenging at present because the microcosmic interaction between the polarization directions is difficult to observe directly. The experimental verification requires complex device design and delicate experimental measurements, especially operation of rotation angle [1,2]. Based on our experience on electronically manipulated magnetic anisotropy [3], we would design new experiments to obtain the mutual regulation between magnetic and electrical anisotropies in our future work. Herein, we have provided adequate analysis and independent control of electrical and magnetic anisotropies.

According to the comments from the reviewer, we have changed our title to “Electrical and Magnetic Anisotropies in van der Waals Multiferroic CuCrP_2S_6 ”, which reflects the point of innovation more clearly.

Reference

- [1] Chiba, D.; Sawicki, M.; Nishitani, Y.; Nakatani, Y.; Matsukura, F.; Ohno, H. Magnetization Vector Manipulation by Electric Fields. *Nature* **455**, 515-518 (2008).
- [2] Matsukura, F.; Tokura, Y.; Ohno, H. Control of Magnetism by Electric Fields. *Nat. Nanotechnol.* **10**, 209-220 (2015).
- [3] X. L. Wang, H. L. Wang, J. L. Ma, X. P. Zhao, and J. H. Zhao, Efficiently rotating the magnetization vector in a magnetic semiconductor via organic molecules, *ACS Appl. Mater. & Interfaces* **11** (6), 6615 (2019).

The rationale for this decision:

First, it is a pretty standard exercise to analyze AFMR in AFM systems with uniaxial anisotropy, although the authors claim to have, for the first time, measured AFMR in the vdW multiferroic.

Reply: We thank the reviewer for this comment. We also appreciate the reviewer for leading us to get a deep understanding and a useful practice to analyze AFMR. Although the analysis of uniaxial anisotropy in FMR is a standard exercise, AFMR in 2D magnets is a new research direction and has attracted much interest recently. In this work, we have utilized the Landau-Lifshitz model to fit for A-type AFM semiconductor, which was just developed several months ago, according to Ref. [43] in the revised manuscript. To be more accurate, we stated that “the spin-flop transition of Néel vector accompanied with highly consistent resonance modes are intriguingly discovered and adequately analyzed” in the revised manuscript. We did not claim that “to have, for the first time, measured AFMR in the vdW multiferroic”.

Second, electric and magnetic anisotropy is characterized in a separate temperature window. The sum of individual characterizations is insufficient to make a strong case in multiferroic materials.

Reply: We thank the reviewer for this comment. As we know, it is difficult to measure the in-plane ferroelectric polarization at low temperature by PFM as compared to out-of-plane polarization. Meanwhile, CCPS is quite insulating at low temperature and it is thus difficult to obtain the current rectifications within the vdW layer. The research hotspot of multiferroic materials is how to control the electric polarization by magnetic field, or manipulate the magnetic polarization by electric field. It is not necessary that the electrical and magnetic anisotropies occur at the same temperature. As we have demonstrated the magnetoelectric coupling effect, one can, in principle, design the proper device to realize the magnetization vector manipulation by electric fields in the magnetic temperature window, or polarization vector modulation by magnetic field in the electric temperature window. From a broader perspective, multiferroic materials can be applied to many aspects. The electrical and magnetic anisotropies can be tuned by external stress, electrostatic doping, interface effects, etc. Therefore, the individual characterizations and systematical analysis of this emerging 2D vdW multiferroic are of great importance for novel device design.

Reviewer #2 (Remarks to the Author):

The authors have addressed all the concerns and revised the manuscript accordingly. Now I would recommend it for publication in Nature Communications.

Reply: We appreciate all the constructive comments of our manuscript from the reviewer.

Reviewer #3 (Remarks to the Author):

The authors have addressed my questions in the response and I find this work suitable to be considered for the publication in Nature Communications.

Reply: We appreciate all the constructive comments of our manuscript from the reviewer.